# Effects of Cr Stress on Bacterial Community Structure and Composition in Rhizosphere Soil of *Iris tectorum* under Different Cultivation Modes

**Zhao Wei [1,2], Zhu Sixi [1,*], Gu Baojing [2], Yang Xiuqing [1], Xia Guodong [1] and Wang Baichun [1]**

[1] The Karst Environmental Geological Hazard Prevention of Key Laboratory of State Ethnic Affairs Commission, College of Eco-Environment Engineering, Guizhou Minzu University, Guiyang 550025, China

[2] College of Environment and Resources Science, Zhejiang University, Hangzhou 310058, China

\* Correspondence: zhusixi2011@163.com

**Abstract:** With the rapid development of industry, Cr has become one of the major heavy metal pollutants in soil, severely impacting soil microecology, among which rhizosphere microorganisms can improve the soil microenvironment to promote plant growth. However, how rhizosphere bacterial communities respond to Cr stress under different cultivation modes remains to be further studied. Therefore, in this study, a greenhouse pot experiment combined with 16S rRNA high-throughput sequencing technology was used to study the effects of Cr stress at 200 mg kg$^{-1}$ on the bacterial community structure and diversity in the rhizosphere soil of *Iris tectorum* under different cultivation modes. The results showed that the rhizosphere bacterial community diversity index (Shannon and Simpson) and abundance index (Ace and Chao) increased significantly with wetland plant diversity under Cr stress. Moreover, the bacterial community changed by 20.1% due to the addition of Cr, further leading to a 15.9% decrease in the common species of the bacterial community, among which *Proteobacteria*, *Actinobacteria*, *Chloroflexi*, and *Acidobacteriota* accounted for more than 74.8% of the total sequence. However, with the increase in plant diversity, the abundance of rhizosphere-dominant bacteria and plant growth-promoting bacteria communities increased significantly. Meanwhile, the symbiotic network analysis found that under the two cultivation modes, the synergistic effect between the dominant bacteria was significantly enhanced, and the soil microenvironment was improved. In addition, through redundancy analysis, it was found that C, N, and P nutrients in uncontaminated soil were the main driving factors of bacterial community succession in the rhizosphere of *I. tectorum*, and Cr content in contaminated soil was the main driving factor of bacterial community succession in *I. tectorum* rhizosphere. In summary, the results of this study will provide a basis for the response of the rhizosphere bacterial community to Cr and the interaction between wetland plants and rhizosphere bacteria in the heavy metal restoration of wetland plants under different cultivation modes.

**Keywords:** Cr stress; rhizosphere bacterial community; *Iris tectorum*; 16S rRNA sequencing technology; phytoremediation

## 1. Introduction

With the influence of human activities, chromium (Cr) has been widely used in tanning, battery manufacturing, fertilizer, textile, carpet, and electroplating industries [1]. This makes it the most common metal contaminant in groundwater, soil, and sediment. Hexavalent $Cr^{6+}$ and trivalent $Cr^{3+}$ are the two most stable and common forms of Cr. Compared with $Cr^{3+}$, $Cr^{6+}$ is more toxic to plants and microorganisms because of its higher solubility, mobility, and oxidation capacity [2]. It can damage the genetic material of all living things and is currently classified as an A class carcinogen by the United States Environmental Protection Agency (USEPA) [3]. Studies have shown that heavy metals can inhibit plant growth and, thus, photosynthesis [4], destroy protein structure, and even change cell morphology [5,6]. Therefore, reducing the toxicity of Cr to the environment

has become a concern of people. As for the traditional heavy metal removal methods, bioremediation has the advantages of less investment, low operating cost, and no secondary pollution. Therefore, bioremediation is widely used in $Cr^{6+}$-contaminated soil and water, mainly in removing $Cr^{6+}$ through adsorption, bioaccumulation, and biotransformation [7]. Among them, the rhizosphere is the leading site of Cr migration and transformation, where plants promote the development of bacterial populations by releasing root exudates [6]. Moreover, the rhizosphere provides a variety of carbon-rich microhabitats that can provide a favorable environment for colonizing beneficial bacterial people [8,9].

In recent years, with the rapid development of high throughput sequencing technology, plant rhizosphere microflora research has achieved remarkable progress [10,11], mainly reflected in plant rhizosphere microbial screening, the identification of genes that have heavy metal tolerance, tolerance and the succession of microbial community characteristics and driving factors analysis, and artificial wetland biological strengthening technology [12,13]. The study found that plant growth-promoting bacteria (PGPB) under Cr stress can promote plant growth through mineral phosphate solubilization, nitrogen fixation, indole-3-acetic acid, siderophores, hydrogen cyanide, and ammonia production [14]. At the same time, the strains with Cr tolerance were screened out in previous studies. They are, respectively, *Serratia sp*. and *Arthrobacter* sp. [15], (*Pseudomonas Alcaliphila Strain* NewG-2) [16], (*Pseudomonas Strain* CPSB21), *Bacillus* cereus. They can alleviate Cr stress by their stress mechanism and promote plant growth. Furthermore, the removal rate of $Cr^{6+}$ by Bacillus cereus was close to 100% when pH = 7 and the temperature was 35 °C [17]. Therefore, under the background of extensive research on rhizosphere microorganisms, it is of great significance to explore the response of the rhizosphere microbial community to Cr stress and the protection of microorganisms on plants.

Constructed wetlands have been widely used in treating Cr-containing wastewater because of their ecological and environmental advantages and economic and social benefits. Substrates and microorganisms fix more than 90% of Cr removal by constructed wetlands, and less than 10% of Cr is directly removed by plants, indicating that substrates and microorganisms are the keys to Cr removal by constructed wetlands [17,18]. Wetland plants can fix Cr in the cell wall, cell membrane, and vacuole through their stress matrix and exist in chelates and complexes, thus reducing Cr toxicity [6,19]. *I. tectorum* is a perennial herbaceous species with developed roots, strong stems, and rapid growth. At the same time, it has strong adaptability to the extreme living environment and has strong tolerance and enrichment of heavy metals [20]. Therefore, *I. tectorum* can serve as a promising material base carrier under Cr stress and, at the same time, combine with beneficial microorganisms to repair Cr-contaminated soil and water [21]. However, most current studies focus on the stress response of rhizosphere bacterial communities to Cr stress under monoculture mode, and further studies are needed under different cultivation modes. Therefore, Taking *I. tectorum* as the research object, 16S rRNA high-throughput sequencing technology was used to analyze the effects of Cr pollution on the structure, composition, diversity, and symbiotic network of soil microbial community under different cultivation modes and to explore the effects of Cr pollution on soil microbial community characteristics under different cultivation modes. To reveal the symbiotic network pattern of microbial communities in Cr-contaminated soil under different cultivation modes. This study provides a theoretical basis for further understanding chromium stress's effects on soil microbial community composition and bioremediation of chrome-contaminated soil under different cultivation modes.

## 2. Materials and Methods

### 2.1. Experimental Design

The background value of heavy metal Cr in Guizhou Province is 95.9 mg kg$^{-1}$ [22]. The Cr content of farmland soil should be controlled below 350 PPM [23]. This study refers to the study of Wang et al. [24] and takes the intermediate concentration (200 mg kg$^{-1}$) as the Cr pollution concentration in our experiment. The *I. tectorum* seedlings used in the

experiment were cultivated by the Plant Seedling company through strict hydroponics (the plants were suspended from the plate, part of the root system was inserted into the nutrient solution, and part of the root system was grown in the space between the nutrient solution and the plate). Firstly, after screening *I. tectorum* seeds of the same size, surface disinfection was carried out with 75% alcohol. The seeds were then washed five times with deionized water and placed in a damp petri dish to germinate. The germinated seeds were moved to the colonization plate and cultured with a nutrient solution. When they grew to 20–30 cm, we purchased these seedlings from the company and selected them with the same growth for the experiment. The seedlings were surface-disinfected with 75% ethanol and 1% sodium hypochlorite for 10 s and 15 min [25], carefully washed with deionized water five times, and then transplanted into potted greenhouse plants with three plants in each pot. A 1/4 concentration of 100 mL of Hoagland solution was added to each pool once a week during cultivation to ensure the nutrients needed for *I. tectorum* growth. Water depth was controlled with deionized water.

Experiments were performed in 32 × 28 cm (diameter × height) pots. *I. tectorum* was cultivated by a greenhouse pot experiment in a flooded condition. Each pot contained 20 kg of soil collected from a karst mountain in southwestern China (106°37′36″ E, 26°22′26″ N), and multi-point mixed sampling was conducted to take soil samples from depths of 0–20 cm. Weed stones were removed and passed through a 2-mm sieve. The pot of sole-cultivated pattern (ITI), two-cultivated pattern (ITII), and three-cultivated pattern (ITIII) was designed as the control group (original soil samples without Cr contaminated), contaminated group (the exogenous addition of 0.1 mmol L-1 $K_2Cr_2O_7$ solutions made Cr(VI) content to be 200 mg $kg^{-1}$ in the soil), and the un-planted blank samples (CK), respectively. Each group had three replicates. Details of the experimental grouping arrangement are shown in Figure S1. The greenhouse ensured a temperature of 25 °C and moisture content of 50%, reducing micro-meteorological factors' interference with plant growth.

## 2.2. Sampling and Chemical Analysis

The pot culture started from the seedling (20–30 cm), and Cr(VI) was added to the pollution group after 30 days of domestication. After three months of the pot experiment, destructive samples were taken. Randomly selected plants in potted plants were removed from the soil, and the ground attached to the primary root was collected (served as rhizosphere soil) and refrigerated to preserve the selected plants [26]; CK groups were potted to take soil samples of 0–5 cm from the upper layer. All models were taken three times each time, totaling 21 samples. Finally, samples were frozen at −20 °C for DNA extraction, high throughput sequencing, and determination of Cr content [6].

After harvesting the rhizosphere soil, all other soil samples were placed on yellow paper and air dried at the air outlet. After air drying, soil samples were repeatedly knocked in a cloth bag, screened with 100 mesh, and then put into a separate sealed bag for subsequent determination of pH, SOM, TN, TP, $NH_4^+$-N, and $NO_3^-$-N. The Cr content of uncontaminated soil was measured by atomic absorption spectrophotometer (Perkin Elmer Analyst 800, USA). A pH meter was used to determine soil pH. Soil organic matter (SOM) was measured by $K_2Cr_2O_7$-$H_2SO_4$ oxidation–external heating method [27]; soil total nitrogen (TN) was measured by the Kjeldahl method of nitrogen determination [28]; and soil ammonium nitrogen ($NH_4^+$-N) and soil nitrate nitrogen ($NO_3^-$-N) were analyzed using 1 M potassium chloride (KCl). Sodium salicylate and hydrazine sulfate were added to produce a color reaction, and absorbance was measured spectrophotometrically at 660 nm and 550 nm [6]. Soil total phosphorus (TP) was measured by $H_2SO_4$ digest-Mo-Sb anti spectrophotometer method [29]. The Cr concentration in plants was determined by atomic absorption spectrophotometry [30].

## 2.3. DNA Extraction, 16S rRNA Gene Sequencing, and Shotgun Metagenome Sequencing

Total genomic DNA from the IT rhizosphere and bulk soil was extracted using a FastDNA SPIN Kit for Soil (Qbiogene Inc., Carlsbad, CA, USA) following the manufac-

turer's instructions and sent to Majorbio Bio-pharm Technology, Shanghai, for sequencing. Sequencing interval: 338F 806R. The primer's name was 338F/806R. Primer sequences: ACTCCTACGGGAGGCAGCAG/GGACTACHVG GGTWTCTAAT. Samples were randomly selected for pre-experiments to ensure that the majority of samples in the lowest number of cycles could be amplified to the appropriate concentration of products. After the preliminary experiment, TransGen ap 221-02 was used in the formal PCR test: TransStart Fastpfu DNA Polymerase, 20 μL reaction system: Table S1. Using PCR instrument: ABI GeneAmp[®] 9700, PCR reaction parameters: a. $1 \times (3$ min at 95 °C), b. cycle number $\times (30$ s at 95 °C; 30 s at annealing temperature 80 °C; 45 s at 72 °C), and c. 10 min at 72 °C, 10 °C until halted by the user. PCR products are detected by 2% agarose gel electrophoresis. The Major Cloud platform was used to conduct OTU cluster analysis of sequencing results (based on Research software). Sequences were optimized to extract non-repeating sequences, and sequences with more than 97% similarity with representative sequences were selected. The OTU number was calculated (that is, the observed abundance sobs value), and Alpha diversity analysis was conducted to analyze species richness and diversity [6].

### 2.4. Statistical Analysis

Shapiro–Wilk and Levene's tests tested the normality and homogeneity of data. The Kruskal–Wallis test was used to analyze the soil's physicochemical properties and diversity index. All microbial visualizations (PCA, Veen, symbiotic network, species composition, etc.) were made using the online platform (www.majorbio.com; 12 August 2022). SPSS 26.0 statistical software was used for all statistical analysis, Origin 2021 was used for all histograms and boxplots, and Adobe Illustrator CC 2019 was used for schematics.

## 3. Results

### 3.1. Effects of Cr stress on Soil Physicochemical Properties and Heavy Metal Accumulation in Plants

In this study, compared with the CK group, soil pH and SOM contents decreased significantly after inoculating wetland plants and showed an overall downward trend with the increase of plant species. However, under the three planting modes, the addition of exogenous Cr significantly increased pH and SOM contents (Figure 1a,b). In addition, with the increase of plant species, the contents of TP, TN, and $NH_4^+$-N in soil showed a decreasing trend. However, they showed a significant increasing trend except for TN after Cr(VI) addition, and the increase was the most significant under the three modes ($p < 0.05$) (Figure 1c–f). Meanwhile, with the addition of exogenous Cr, the content of heavy metals in plant roots, stems, and leaves increased significantly ($p < 0.05$), and the increase of Cr content in the single planting mode was the most obvious, which was 74.53%, 67.11%, and 65.85% higher than that in the unpolluted group, respectively. With the increase of plant species, Cr content in the rhizomes and leaves of *I. tectorum* showed an overall decreasing trend (Figure 1g–i). In conclusion, under Cr stress, *I. tectorum* showed good tolerance and enrichment ability to Cr and increased with the increase of total plant classes.

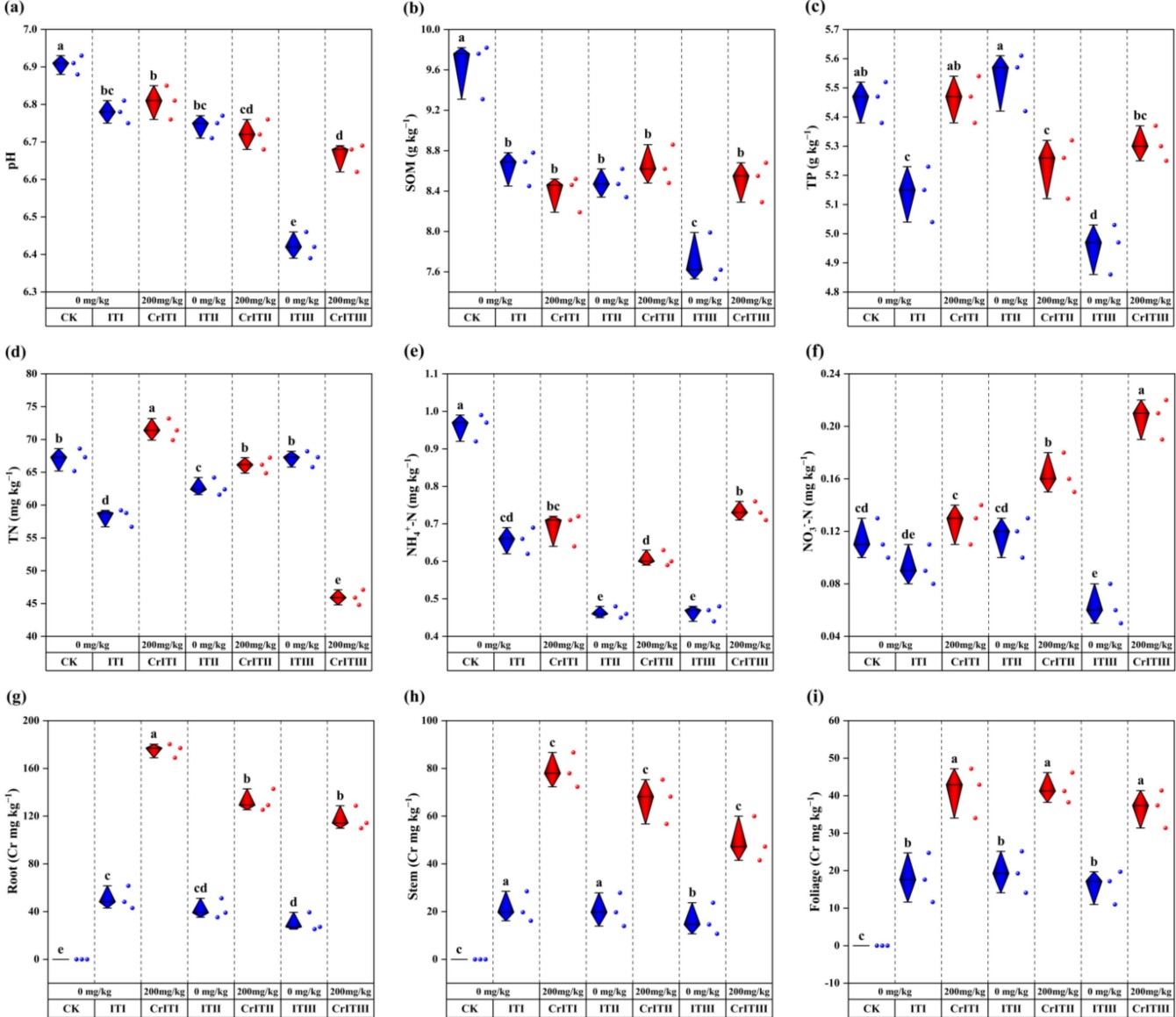

**Figure 1.** *I. tectorum* heavy metal content and soil physicochemical properties under Cr stress ((**a**–**i**): pH, SOM, TP, TN, NH$_4$$^+$-N, NO$_3$$^-$-N, Cr content in roots, foliage, and stems). Data within the same followed by a string of the same lowercase letters are not significantly different ($p > 0.05$). At the same time, a series of other letters show a significant difference ($p < 0.05$).

### 3.2. Rhizosphere Bacterial Diversity Index

A total of 1,239,259 valid sequences were obtained, and 3008 bacterial taxa (OTUs) were identified by high-throughput sequencing technology after removing unrecognized gene base sequences and chimera. The coverage of OTU sequences was above 98%, and the slopes of the dilution curves of all samples were close to saturation (Figure S2). Compared with the CK group, plant inoculation significantly increased the diversity and abundance of the soil bacterial community. With the increase of plant species, the diversity and abundance index increased significantly (Figure 2a–d). After the addition of Cr(VI), the Shannon index, Simpson index, Ace index, and Chao index of the rhizosphere bacterial community were significantly decreased except in single cropping mode ($p < 0.05$). These results indicated that adding exogenous Cr significantly reduced the diversity and abundance of bacterial communities in rhizosphere soil.

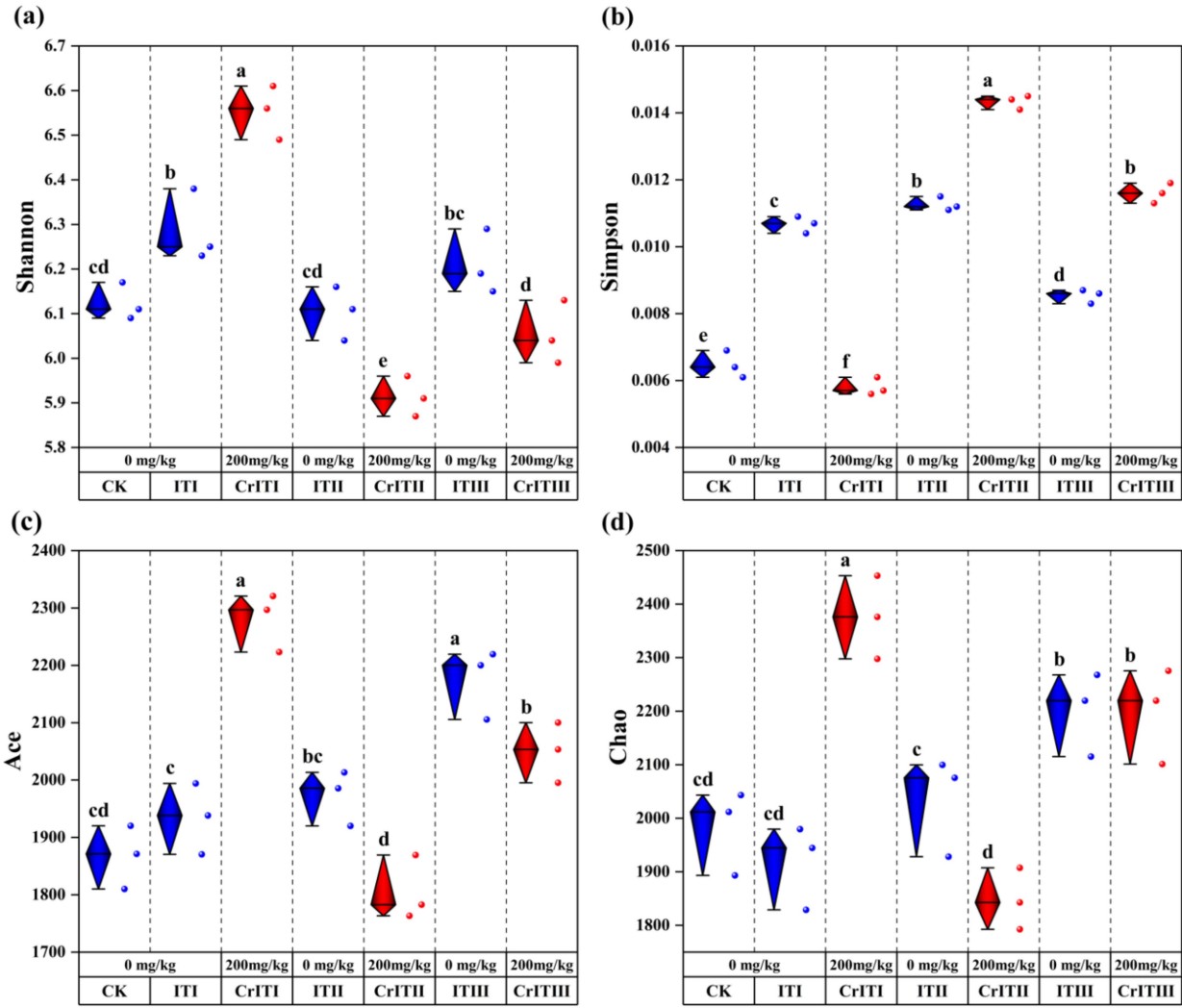

**Figure 2.** Rhizosphere microbial community diversity and abundance index of iris under Cr stress ((**a**) Shannon index; (**b**) Simpson index; (**c**) Chao index; (**d**) Ace index). Data within the same followed by a string of the same lowercase letters are not significantly different ($p > 0.05$). At the same time, a series of other letters show a significant difference ($p < 0.05$).

*3.3. Bacterial Community Structure in Rhizosphere Soil*

In this study, it can be seen from the Venn diagram that the rhizosphere bacterial community composition in the control group and the pollution group had significant changes, among which the total number of species in the control group and the pollution group accounted for 49.8% and 41.9%, respectively (Figure 3e,f). To further understand the differences in bacterial community structure under Cr pollution, (PCoA) and (NMDS) were conducted on the OTU level for each group of samples to explore the spatial and temporal distribution pattern of the bacterial community in soil (Figure 3a,b). The bacterial community structure was significantly different between the control and the contaminated groups. The similarity analysis of all samples showed that Cr pollution greatly impacted soil bacterial community structure (R = 0.1623, $p < 0.05$, Figure 3d). PLS-DA analysis was carried out to prove the contribution of Cr stress on the change of the *I. tectorum* rhizosphere bacterial community (Figure 3c), in which the samples were divided into control and pollution groups. The results showed that the bacterial community changed by 20.1% due to Cr stress, leading to the control and the contaminated groups showing two apparent divisions. Therefore, exogenous Cr supplementation can significantly affect bacterial communities' spatial patterns and composition.

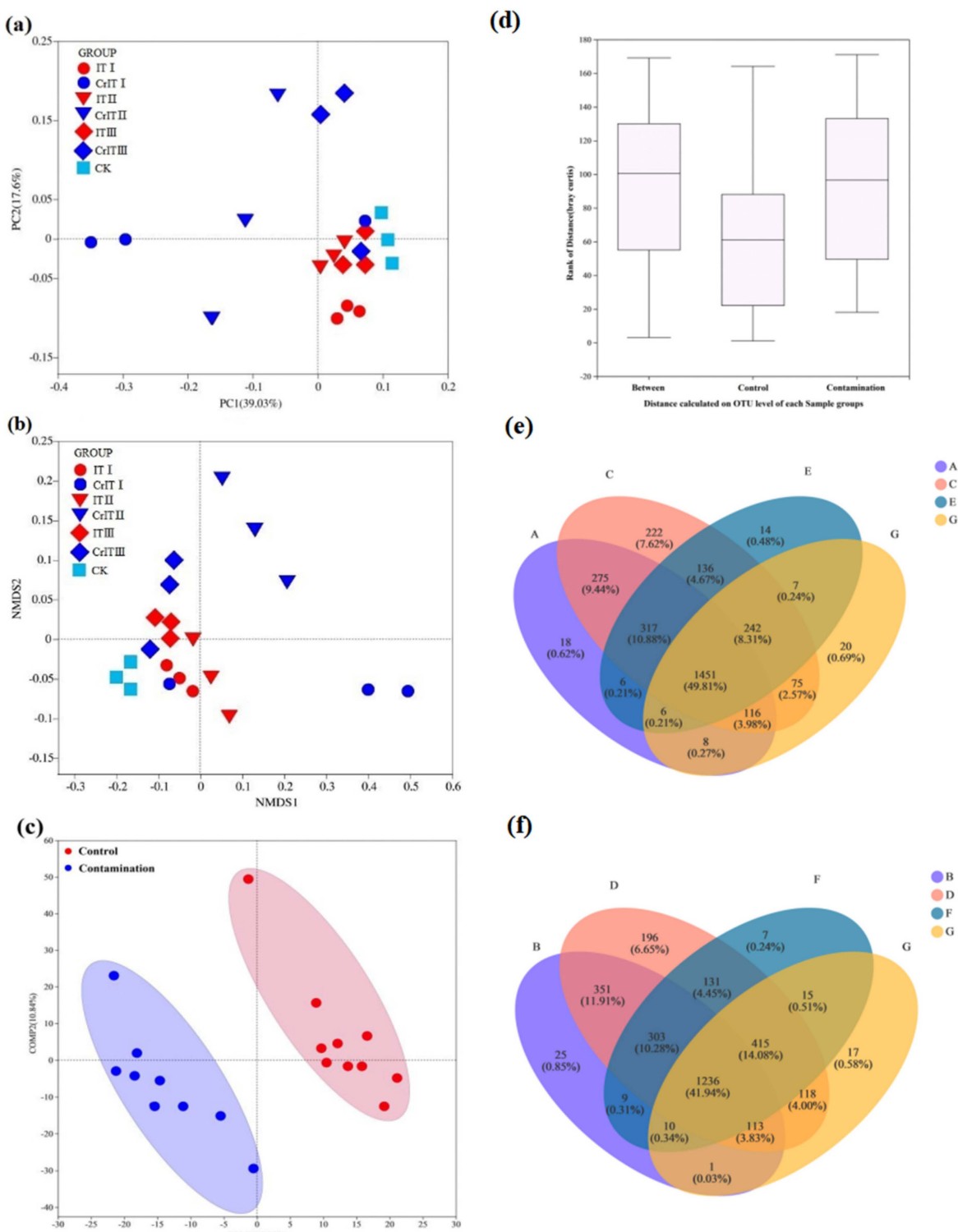

**Figure 3.** PCoA analysis on weighted UniFrac distance (**a**), NMDS analysis on Bray-Curtis distance (**b**), PLS-DA analysis (**c**). ANOSIM analysis (R = 0.1623, *p* < 0. 05) (**d**). The control contains treatments of ITI, ITII, and ITIII (**e**), and the Contamination contains treatments of CrITI, CrITII, and CrITIII. I and (**f**) are Venn diagrams of rhizosphere microbial communities in the control and pollution groups. Different colors represent different treatments. The overlapped part is the number of common species, while the non-overlapped part is the number of unique species. A, C, and E were the control group without Cr in *I. tectorum* single, double, and three plants, respectively. B, D, and F were polluted groups with *I. tectorum* single, double, and triple Cr treatments, respectively. G is for unplanted and bulk soil (CK).

### 3.4. Bacterial Community Composition in Rhizosphere Soil

Our study identified 3008 bacterial taxa belonging to 36 species, 90 classes, and 830 genera. The dominant phyla were Proteobacteria (average relative abundance 38.7%), Actinobacteria (17.5%), Chloroflexi (10.0%), Acidobacteria (8.6%), Gemmatimonadetes (7.7%), Bacteroidetes (4.3%), and Firmicutes (4.1%) (Figure 4a). In the control group, Proteobacteria showed an apparent upward trend under the three-cultivated pattern. Actinobacteria and Chloroflexi have the highest abundance in the sole-cultivated design (even higher than in the blank group). Acidobacteria and Gemmatimonadetes had the highest surplus in the three-cultivated pattern, whereas they were lower than the open group. However, compared with the control group, the abundance of Proteobacteria increased significantly under the two-cultivated pattern and three-cultivated pattern after Cr addition. The variation trend of Actinobacteria and Chloroflexi was consistent with that of the control group, but their abundance was slightly lower than that of the control group. Acidobacteria and Gemmatimonadetes also have the highest quantity in the three-cultivated pattern, but the mass is still lower than that in the control group. These results showed that the dominant bacteria were not significantly affected under Cr stress. In contrast, some less abundant bacteria (such as Bacteroidetes, Firmicutes, and Patescibacteria et al.) fluctuated considerably, and the bacterial community abundance was significantly improved under a two-cultivated pattern and three-cultivated pattern. At the same time, compared with the CK group, Geobacter abundance in rhizosphere soil was significantly increased after inoculation with wetland plants. However, the *Geobacter* abundance generally declined with increasing plant species. In addition, the abundance of *Gemmatimonas* and *Sphingomonas* decreased significantly after inoculation with wetland plants but increased with increasing plant species (Figure 4b). Based on LDA < 2.5, Lefse analysis was performed to further compare the changes of rhizosphere bacterial communities in the control group and the contaminated group under different cultivation modes. Significant differences in the control group were *Micrococcaceae* (LDA = 4.02), *Chloroflexales* (LDA = 3.60), *Roseiflexaceae* (LDA = 3.51), *Acidobacteriia* (LDA = 3.44), etc. After Cr stress, *Proteobacteria* (LDA = 4.65), *Gammaproteobacteria* (LDA = 4.63), *Xanthomonadales* (LDA = 4.37), *Lysobacter* (LDA = 4.23), *Acidovorax* (LDA = 4.09), and *Rhizobiales* (LDA = 3.83) changed significantly (Figure 5a). Through the construction of an evolutionary tree, the relationship between species in the sample was revealed from the perspective of molecular evolution. According to the ratio of Reads, the main bacteria groups that had significant changes in the control group and the pollution group after inoculation were *Actinobacteria*, *Gemmatimonadetes*, *Patescibacteria*, and *Firmicutes* (Figure S3).

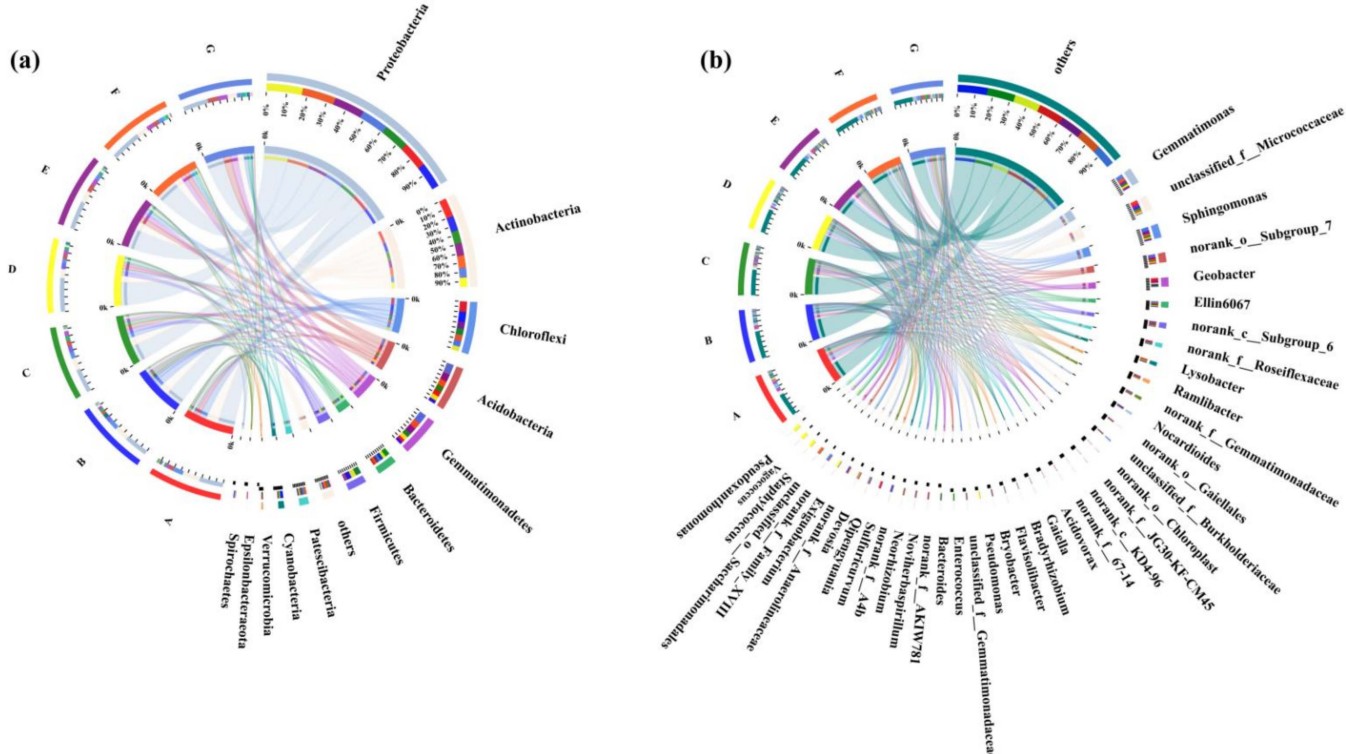

**Figure 4.** Rhizosphere soil microbial community structure and differences of *I. tectorum* under Cr stress under different treatments ((**a**) Phylum level; (**b**) Genus level). The small semicircle (left half circle) represents the species composition in the sample. The color of the outer band represents the group from which the species are. The color of the inner band represents the species, and the length represents the relative abundance of the species in the corresponding sample. The large semicircle (right half circle) represents the distribution ratio of species in different samples at the taxonomic level. The outer band represents species, the inner band color represents different groups, and the length represents the distribution ratio of the sample in a particular species. A and B are *I. tectorum* monoculture control and Contamination, respectively; C and D are *I. tectorum* bio culture control and Contamination, respectively; E and F are the control and Contamination of *I. tectorum*, respectively. G is for unplanted and bulk soil (CK).

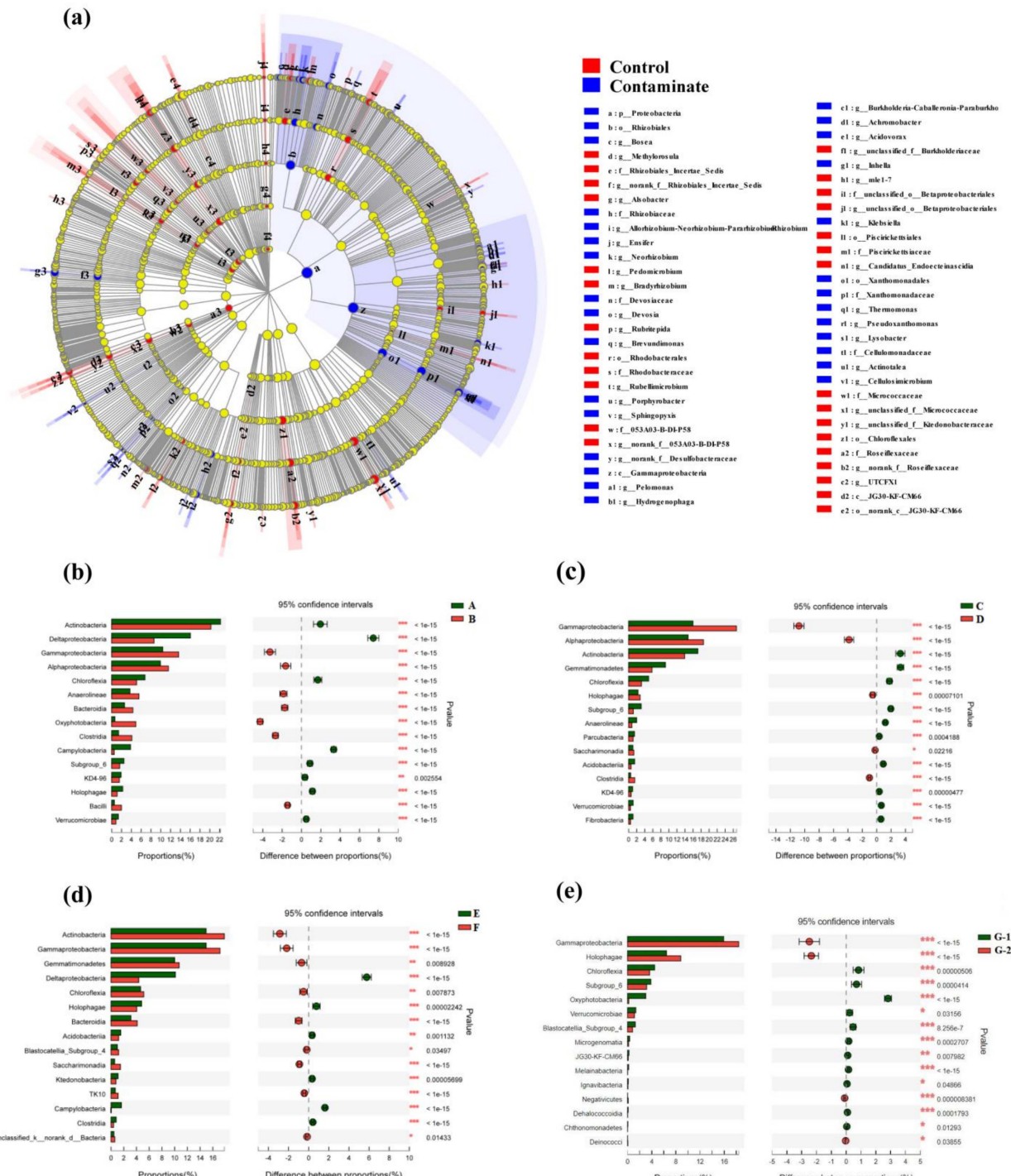

**Figure 5.** Differences of rhizosphere soil microbial community abundance in different cultivation modes of Iris under Cr stress (**a**): Lefse multi-level discriminant analysis of species difference showed that different color nodes represented the microbial groups that were significantly enriched in the corresponding groups and had a significant impact on the differences between groups. The light yellow nodes indicate the microbial groups that were not significantly different in different groups or had no significant effect on the differences between groups. (**b–e**): Wilcoxon rank-sum test bar plot on the class level. Each column corresponding to the species represents the relative abundance of the species in each sample, and different colors represent different samples. The middle region is within

the confidence interval set, and the values corresponding to the dots represent the difference in the relative abundance of species between the two samples. A and B are *I. tectorum* monoculture control and Contamination, respectively; C and D are *I. tectorum* bio culture control and Contamination, respectively; E and F are the control and Contamination of *I. tectorum*, respectively. G is for unplanted and bulk soil (CK). * stands for $p < 0.05$, ** represents $p < 0.01$, *** represents $p < 0.001$.

According to the histogram analysis of species abundance difference (Figure 5b–e), the abundance of Actinobacteria and Deltaproteobacteria was the highest in the sole-cultivated pattern, and the quantity of the polluted group was significantly lower than that of the control group. Gammaproteobacteria and Alphaproteobacteria had the highest abundance under a two-cultivated design, and Cr stress increased their mass. Actinobacteria in the three-cultivated pattern showed an opposite situation to that in the sole-cultivated practice, and the beneficial bacteria population, Gemmatimonadetes, appeared again as the dominant species.

### 3.5. Interspecific Symbiotic Network of Soil Bacterial Communities

The symbiotic network diagram of bacterial communities was used to evaluate the correlation between different bacterial communities, reflect the coexistence pattern of bacterial communities in specific habitats, and explore the interspecific relationship of soil bacterial communities under Cr stress (Figure 6a–c). The colinear network diagram indicated that bacterial community structure among all samples was similar (Figure S4). The dominant species are Proteobacteria, Actinobacteria, Chloroflexi, Acidobacteria, Gemmatimonadetes, and Nitrospinae. Through further analysis, we found that the dominant species in the sole-cultivated pattern are mainly symbiosis, but the competition between the dominant species is still prominent. The symbiosis needs to be concentrated. Proteobacteria, Actinobacteria, and Chloroflexi are primarily divided into three blocks (Figure 6a). Under the three-cultivated pattern, the symbiotic relationship between dominant species is significantly weakened, and the symbiotic system is increasingly dispersed (Figure 6c). Under the two-cultivated design, the symbiotic relationship between dominant species is enhanced considerably, while the competition relationship is significantly reduced. Moreover, the relationship between species is closer, and the symbiotic system is highly concentrated (Figure 6b and Table S2). Therefore, the results show that the symbiotic system formed among dominant species in the two-cultivated pattern is more conducive to resisting Cr stress.

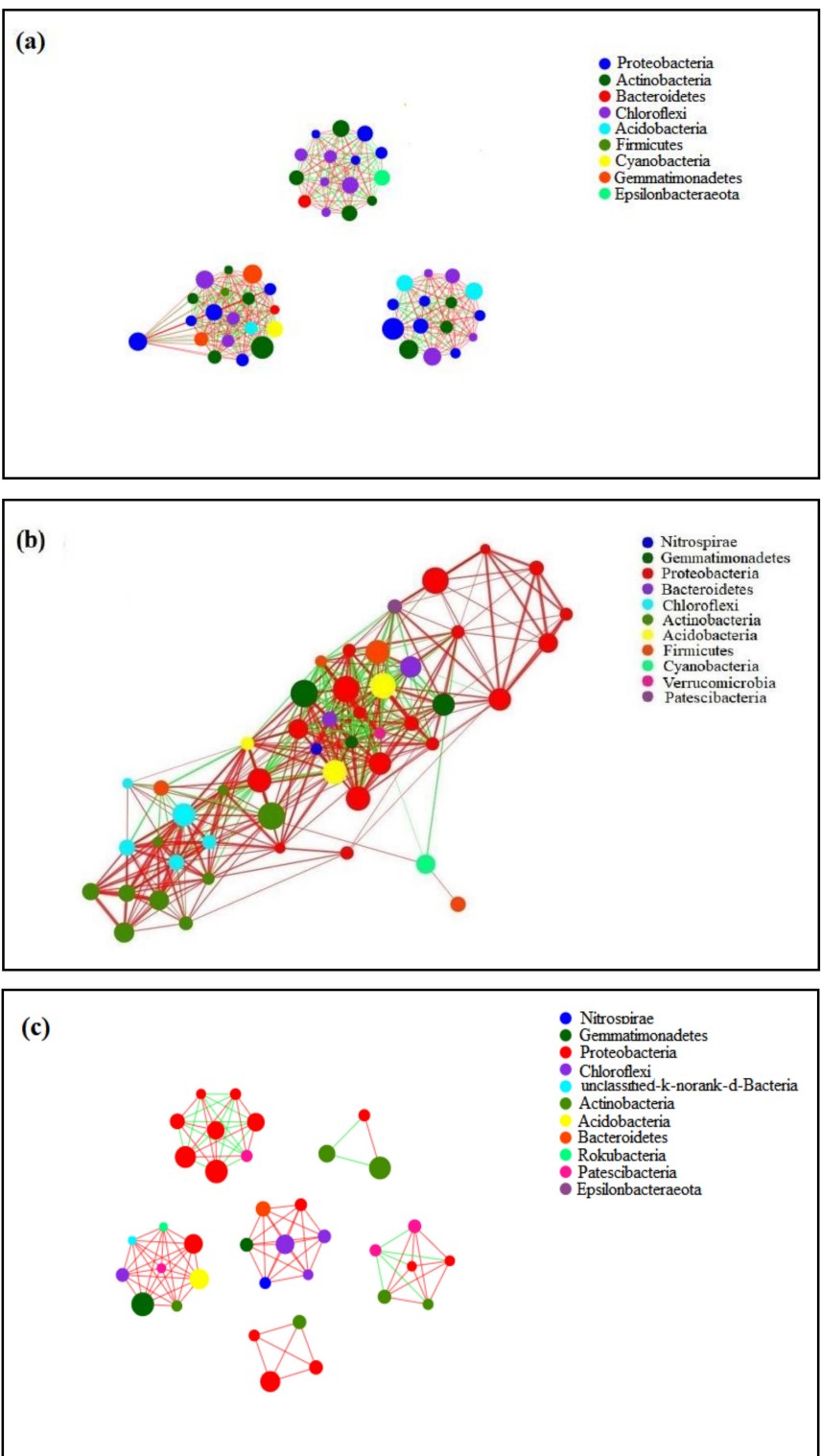

**Figure 6.** Symbiotic network diagram of soil rhizosphere microbial community ((**a**) *I. tectorum* sole-cultivated pattern under Cr stress (**b**) *I. tectorum* two-cultivated pattern under Cr stress; (**c**) *I. tectorum* three-cultivated pattern under Cr stress). The size of nodes in the figure indicates the abundance of species, and different colors indicate different species. The line's color indicates a positive and negative correlation, red indicates a positive correlation, and green indicates a negative correlation. The thickness of the line indicates the correlation coefficient. The thicker the line, the higher the correlation between species. The more lines, the more closely related the species is to other species. B,

D, and F were polluted groups with *I. tectorum* single, double, and triple Cr treatments, respectively. The network diagrams of B, D, and F are (**a**), (**b**), and (**c**), respectively. B represents the network diagram under single planting mode Cr treatment, D represents the network diagram under double planting mode Cr treatment, and F represents the network diagram under three planting mode Cr treatment.

### 3.6. Correlation Analysis between Rhizosphere Soil Bacterial Community and Physicochemical Properties

RDA analysis showed that for the rhizosphere bacterial communities in uncontaminated and contaminated soil, the interpretation rates of the first two principal axes for the total coefficient of variation of *I. tectorum* bacterial communities were 80.34% and 85.78%, respectively (Figure 7). It can be seen from the figure that SOM, TN, $NH_4^+$-N, and pH are the main environmental factors driving the succession of rhizosphere bacterial communities in uncontaminated soil, followed by TP and $NO_3^-$-N. In addition, Proteobacteria, Acidobacteria, and Gemmatimonadetes were positively correlated with TP, TN, $NH_4^+$-N, and SOM and negatively correlated with pH. SOM was the main influencing factor of Actinobacteria; Firmicutes and Patescibacteria were positively correlated with $NO_3^-$-N. After adding Cr(VI), $NO_3^-$-N, SOM, and Cr contents were the main environmental factors driving rhizosphere bacterial community succession, followed by $NH_4^+$-N and pH. Meanwhile, Proteobacteria, Actinobacteria, Firmicutes, and Acidobacteria are the most abundant bacterial groups in polluted soil. Proteobacteria is positively correlated with Cr content and negatively correlated with SOM, TN, TP, $NH_4^+$-N, and pH. AP was the main influencing factor of Firmicutes. In addition, the community structure of the uncontaminated and contaminated groups was similar, but the abundance was significantly different.

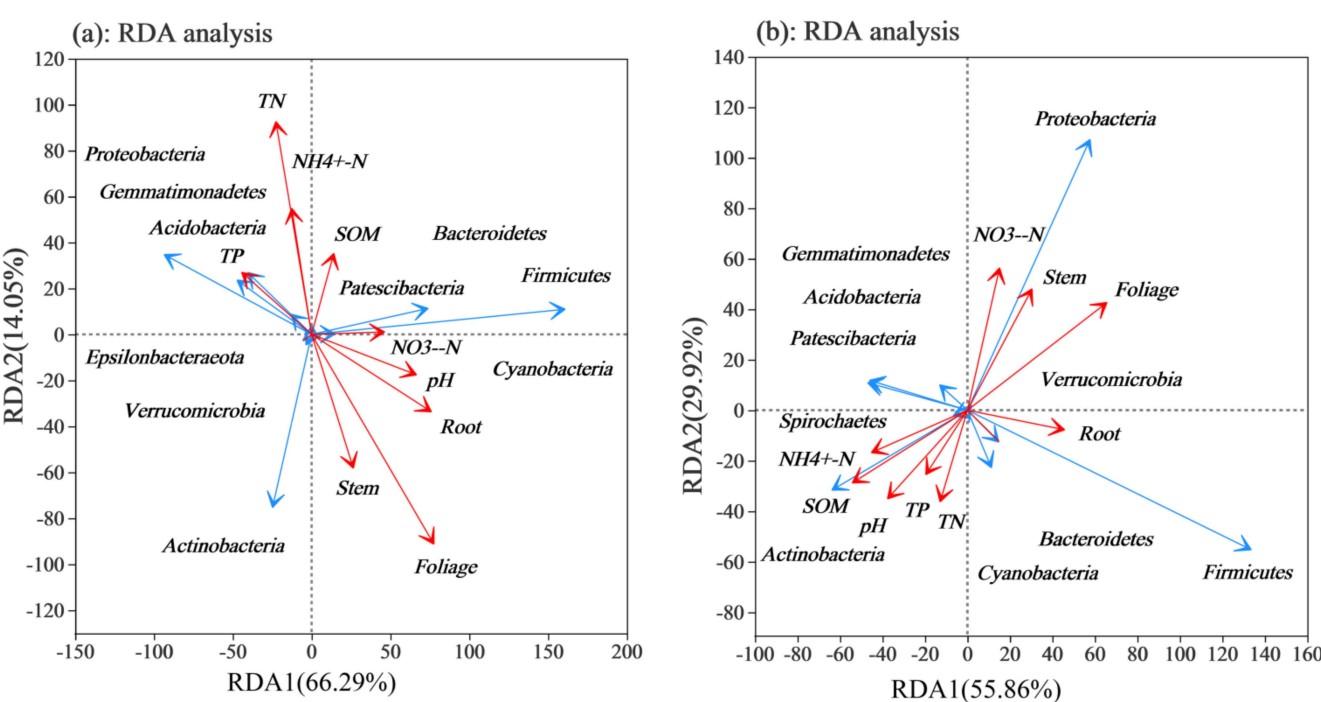

**Figure 7.** RDA analysis of bacterial communities in the rhizosphere of *I. tectorum* ((**a**) uncontaminated soil (**b**) Cr contaminated soil). The red arrow represents quantitative environmental factors, and the length of the environmental factor arrow can represent the impact degree (interpretation quantity) of environmental factors on species data. The angle between the arrows of environmental factors represents positive and negative correlations (acute angle: positive correlation; obtuse angle: negative correlation; right angle: no correlation).

## 4. Discussion

### 4.1. Physiological Response of I. tectorum to Cr Stress

Cr hardly participates in the metabolic function of plants, but it is potentially toxic and can adversely affect its morphophysiological properties and antioxidant defense mechanisms [31]. In the control group, Cr content in roots was significantly higher than that in leaves (Figure 1g–i). When plants grow in Cr-contaminated soil, the seeds are preferentially exposed to heavy metals in the ground, which leads to the accumulation and transport of a large amount of Cr in the hearts of plants [32]. After the exogenous addition of Cr, the content of Cr in the roots and leaves of *I. tectorum* was significantly higher than that in the control group (Figure 1g–i). This may be because wetland plants use their removal mechanism to fix Cr in the cell wall and vacuole of plant cells to purify the soil contaminated by Cr [6]. Moreover, previous studies have shown that hemicellulose (HC-1) in plant cell walls binds heavy metals to the greatest extent under severe mental stress, thus reducing the toxicity of heavy metals to plants [33].

The significant increase in SOM under Cr contamination (Figure 1b) may be due to the following three reasons: (1) Cr would compete with plants for sulfate channels, thus affecting the absorption of water and nutrients by plants [34]; (2) damage to plasma membrane integrity, resulting in damage to root tip cells and more organic matter transport to the soil [35]; (3) dead bacteria in the soil due to the inanimate bacterial biomass accumulating in the soil over time. Meanwhile, Cr stress-sensitive bacterial communities provide the soil with a large amount of organic matter [36]. From the above results, we can see that Cr stress mainly influences plants and rhizosphere bacterial communities, but *I. tectorum* can still enrich Cr in large quantities in extreme environments. Therefore, it can be seen that *I. tectorum* has its advantages in Cr removal.

### 4.2. Effects of Cr Stress on Soil Bacterial Community Structure

Under Cr stress, it is essential to study the complex bacterial community structure and species diversity in plant rhizosphere soil to understand the ecosystem's regulation mechanism [37,38]. In this study, exogenously added Cr reduced the α diversity index (Shannon, Simpson, Ace, and Chao index) of the bacterial community in the rhizosphere of *I. tectorum* (Figure 2). This may be because the high concentration of Cr(VI) inhibited the nitrification and denitrification of the bacterial community, thereby reducing the diversity and abundance index of the bacterial community [39]. Moreover, under the action of reductase or reductive substances, Cr(VI) in cells will produce a large number of reactive oxygen species (ROS) in the process of reduction to Cr(III), which will cause DNA damage after binding with DNA, resulting in cell deformation, genetic variation, and even death [40]. This harms the diversity and abundance index of the bacterial community. In addition, the analysis of PCoA and NMDS showed that Cr stress significantly changed the spatial structure of the bacterial community in the rhizosphere soil of *I. tectorum* (Figure 3a,b), which has also been confirmed in previous studies, such as Windmill Grass [6] and Oryza sativa [41]. In addition, PLS-DA (Figure 3c) and Veen (Figure 3e,f) analysis again proved that exogenous Cr supplementation significantly affected the structural composition and species number of bacterial communities. As seen from the above results, adding exogenous Cr significantly reduces the diversity of the rhizosphere bacterial community of *I. tectorum*. It changes the structure of the rhizosphere bacterial community, causing damage to the macroecology of *I. tectorum*.

### 4.3. Effects of Cr Stress on Rhizosphere Bacteria

Bioremediation relies on the universality and diversity of bacterial communities and responds to Cr stress through its REDOX system, extracellular adsorption, and efflux mechanisms [40,42]. In this study, it was found that Proteobacteria, Actinobacteria, and Chloroflexi were significantly more abundant in the control group than in the blank group (CK) (Figure 4), which may be because root exudates (polysaccharides, phenolic compounds, flavonoids, and organic acids) provide more nutrients for the survival of the

bacterial community (Figure 8), thus promoting the growth of bacteria [43]. Under Cr stress, compared with the control group, the development of Proteobacteria was not inhibited but showed an upward trend (Figure 4). This may be because Proteobacteria are Gram-negative bacteria, and the Gram-negative cell envelope is composed of an outer membrane (containing anion lipopolysaccharide, phospholipid, and outer membrane protein) and peptidoglycan, which plays a crucial role in heavy metal binding [44]. Moreover, bacteria possess polysaccharide slime layers, which readily offer amino, carboxyl, phosphate, and sulfate groups for metal binding [45]. Compared with the control group, the growth of Actinobacteria, Chloroflexi, Acidobacteria, and Gemmatimonadetes was restricted under Cr stress (Figure 4). However, these four dominant bacteria can alleviate heavy metal stress through strong secondary metabolism, more metabolic functions, and self-regulation of energy supply [46]. Actinobacteria, as a typical Gram-positive bacterium, exhibits remarkable Cr resistance, which plays a direct role in reducing and removing Cr [47].

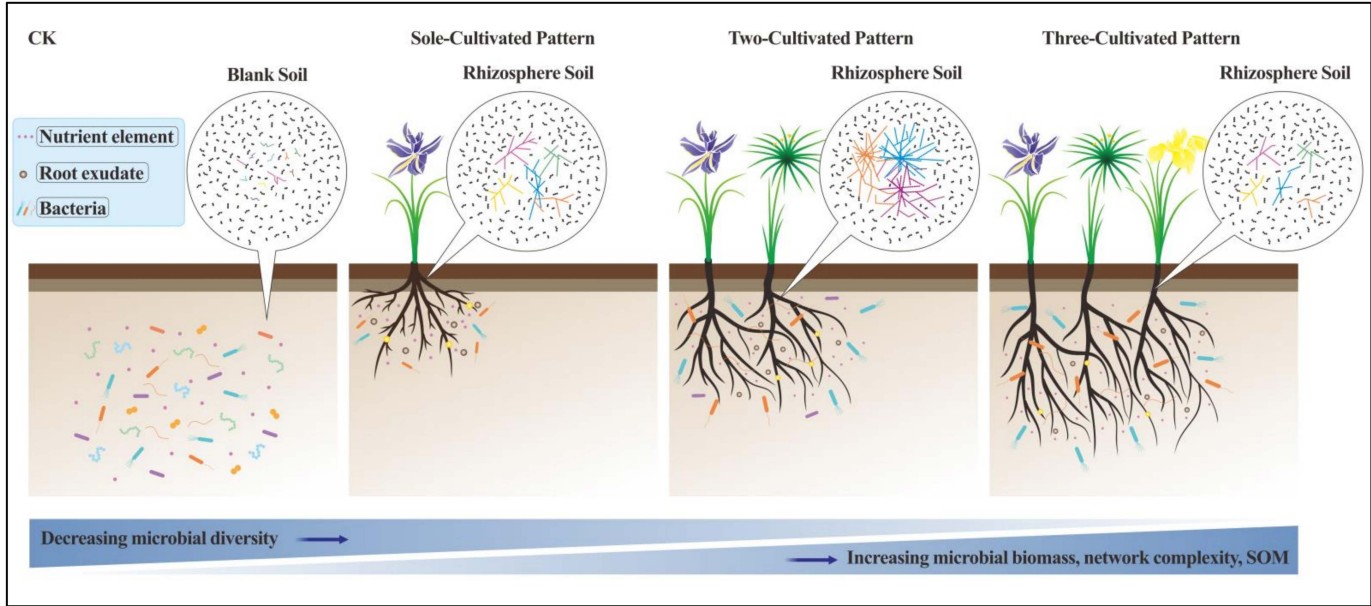

**Figure 8.** Response mechanism of rhizosphere bacterial community under different cultivation modes. Blank soil has a uniform bacterial community without external interference. Sole-cultivated pattern: During plant growth, organic substances are secreted by the root system to make bacterial communities gather around the root system. Moreover, due to the addition of exogenous Cr, a symbiotic network is established among rhizosphere bacterial communities to resist Cr stress jointly. Under a two-cultivated pattern, the increase of plants can significantly improve the soil microenvironment, increase the rhizosphere bacterial community biomass, and build a complete symbiotic network. Under the three-cultivated pattern, the soil microenvironment is improved, but the symbiotic relationship between bacteria is weakened. The effects of different cultivation patterns on the soil microenvironment showed that diversity decreased, biomass increased, network complexity increased, and soil organic matter increased, thus effectively improving the soil microenvironment.

### 4.4. Effect of Different Cultivation Patterns on Rhizosphere Bacteria

　　Through species analysis between groups, it can be seen that Actinobacteria is the dominant species in the sole-cultivated pattern. Still, its abundance is lower than that of the control group. For two-cultivated and three-cultivated practices, not only was the quantity for the bunch of Gammaproteobacteria, Gemmatimonadetes, Alphaproteobacteria, and Actinobacteria highest, but it was also significantly higher than that of the control group. (Figure 5). This may be due to the redistribution of bacterial communities in the soil caused by human disturbances (i.e., different cultivation patterns) or root exudates of wetland plants (Figure 8; [48]). According to the symbiotic network analysis, although

the bacterial community structure among all samples is similar, the correlation between dominant bacterial communities is weak in the sole-cultivated and three-cultivated patterns, and the symbiosis pattern is relatively scattered (Figure 6a,c and Table S2). This may be caused by the sole-cultivated design. The plant itself and its rhizosphere bacterial community cannot cope with Cr stress. Many plant roots and their exudates compete for soil microbial resources in the three-cultivated pattern. This results in the rhizosphere bacterial community's dispersion, which weakens the symbiotic relationship between bacteria. In the two-cultivated practice, the correlation between dominant species is significantly enhanced, the relationship between species is closer, and the symbiotic system is highly concentrated (Figure 6b). It was mentioned in the study that environmental changes played a dominant role in the formation of plant rhizosphere bacterial communities [49,50].

### 4.5. Effects of Soil Physicochemical Properties on Rhizosphere Bacterial Communities

Previous studies have shown that the growth and development of rhizosphere microbial communities can be affected by soil's physical and chemical properties [6]. In this study, we found that C, N, and P elements were the primary driving factors of the rhizosphere bacterial community in uncontaminated soil, which may be because the rhizosphere bacterial community would transform and utilize soil nutrient elements in everyday life activities, to maintain microecological soil balance and its life activities [11]. In addition, during the utilization of the N element by nitrifying and denitrifying bacteria, the intermediate products produced will significantly change the contents of $NO_3^-$-N and $NH_4^+$-N, and $H^+$ will also be released in this process, thus leading to the change of pH in the rhizosphere soil of *I. tectorum* [51]. After adding Cr(VI), $NO_3^-$-N, SOM, and Cr contents were the main environmental factors driving rhizosphere bacterial community succession. At the same time, it is also related to the stress mechanism of the rhizosphere bacterial community [6]. Previous studies have shown that the detoxification of Cr by the bacterial community is mainly through intracellular or extracellular binding and the reduction of heavy metals to reduce their toxicity. At the same time, it can activate its stress mechanism and adjust its nutrient recycling mode to survive better [52]. This study found that Proteobacteria became the most abundant bacterial group in contaminated soil after adding Cr(VI) and showed a significant positive correlation with Cr content (Figure 7). This may be because Proteobacteria has high biotransformation efficiency for Cr(VI), which can reduce Cr(VI) to Cr(III) with low toxicity, and significantly reduces Cr mobility and bioavailability in the environment [53]. The content of SOM is an essential indicator of soil fertility and a necessary substance for the survival of rhizosphere microorganisms. It has been found in this study that Actinobacteria, Firmicutes, and Acidobacteria can survive well under Cr stress and become the dominant bacteria in polluted soil (Figure 7). This may be due to their ability to adjust their nutrient use in extreme environments, where Gram-positive bacteria such as Actinobacteria can sustain their survival through robust secondary metabolism [54], and Firmicutes and Acidobacteria can reduce their growth restriction by actively regulating their energy supply forms and using C, N, and P elements in extreme environments [55,56]. Besides that, pH also plays an essential role in the growth of rhizosphere bacterial communities. For example, the removal rate of $Cr^{6+}$ by Bacillus cereus was close to 100% when pH = 7 and the temperature was 35 °C [17]. The results showed that soil physicochemical properties could drive the succession of rhizosphere bacterial communities. The dominant rhizosphere bacteria could also improve the microenvironment by changing nutrient recycling patterns.

### 5. Conclusions

This study studied the effects of Cr stress on the rhizosphere microenvironment of *I. tectorum* under different cultivation modes. In conclusion, the addition of exogenous Cr significantly reduced the abundance and diversity index of the bacterial community in the rhizosphere soil of *I. tectorum*, significantly changed the spatial pattern of the bacterial community, and affected the composition of the rhizosphere bacterial community. At the

same time, the inoculation of wetland plants changed the composition of the bacterial community in the soil, alleviating the toxic effect of Cr(VI) on the microenvironment. With the increase in plant diversity, the abundance of dominant bacteria in the soil microenvironment increased significantly, and beneficial bacteria for plant growth began to appear successively. Moreover, the two-cultivated pattern effectively changes the symbiosis relationship between dominant species, strengthens the synergy between dominant flora, and forms a more robust symbiosis network. Through RDA analysis, we also found that C, N, and P nutrient elements and Cr content were the primary driving factors for shaping the structure and diversity of the bacterial community in the rhizosphere of *I. tectorum*, and the response was more robust after Cr(VI) was added. The results of this study are helpful to understand further the effects of Cr stress on rhizosphere bacterial communities under different cultivation modes to provide the theoretical basis for the biological remediation of Cr-contaminated soil.

**Supplementary Materials:** The following supporting information can be downloaded at: https://www.mdpi.com/article/10.3390/microbiolres14010020/s1, Table S1—20 μL reaction system. Figure S1—*I. tectorum* pot experiment design. Table S2—Attribute table of bacterial community network under Cr stress cultivation mode. Figure S1—I. tectorum pot experiment design. Figure S2—Bacterial community dilution curve of soil sample. Figure S3—Evolution tree of rhizosphere microbial community of I. tectorum. Figure S4—Collinear network graph.

**Author Contributions:** Software, X.G.; Investigation, Y.X. and W.B.; Writing—original draft, Z.W.; Writing—review & editing, Z.S. and G.B. All authors have read and agreed to the published version of the manuscript.

**Funding:** This study is financially supported by Open Foundation Project of Key Laboratory, Institute of Environment and Plant Protection, Chinese Academy of Tropical Agricultural Sciences (HZSKFKT202204), the National Natural Science Foundation of China (31560107), and by the Science and Technology Support Project of Guizhou province, China (Guizhou Branch Support (2018)2807).

**Institutional Review Board Statement:** Not applicable.

**Informed Consent Statement:** Not applicable.

**Data Availability Statement:** Not applicable.

**Acknowledgments:** I acknowledge the support of the College of Eco-environment Engineering, Guizhou Minzu University; the Karst Environmental Geological Hazard Prevention of Key Laboratory of State Ethnic Affairs Commission.

**Conflicts of Interest:** The authors declare no conflict of interest.

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
