# Peer review of "Effects of Cr Stress on Bacterial Community Structure and Composition in Rhizosphere Soil of Iris tectorum under Different Cultivation Modes"

_2036-7481, doi:10.3390/microbiolres14010020_

Round 1

Reviewer 1 Report

After reviewing the manuscript critically I have identified some issues in this manuscript which will certainly help to improve the quality of the manuscript. My comments are intended to further improve the manuscript such that it addresses the questions asked properly. I would consider this manuscript for publication if queries are addressed and changes made and resubmitted.

1. Introduction is too long, make it more focused and mention the research gaps and hypothesis of this study.

2. For the normality test of the data Shapiro Wilk test will be a good option.

3. Rather than the logarithmic transformation of data the authors should undertake no-parametric methods for analysis like the Kruskal-Wallis test, and Wilcox test.

4. What was the physiochemical properties of the experimental soil? Mention it in the text in materials methods section.

5. How soil pH was measured with potentiometric titration? This totally wrong. Clarify

5. How the nutrient demand of the crops was satisfied? What was the dose/rate of application of fertilizer?

6. Figure 4 and 5 is very difficult to understand. It will be good to use high resolution plots.

Reviewer 2 Report

I have read your work with great interest and attention, and I am full of admiration for the enormous amount of work you have put into the preparation of this manuscript. I really enjoyed your drawings, which are an excellent documentation of your results. Noteworthy is also the discussion chapter in which you showed excellent knowledge of the above issue, as evidenced by the cited 56 references. The issue you have studied is very important for all organisms living on our Earth, so I will recommend the Editors to accept your work for publication without changes.

Reviewer 3 Report

Point 1:  Line 15- I. tectorum full name when writing first time.

Point 2 :  first is quality of figures are too poor it must should be improved.

Point 3 : The second question is about the availability of the data, whether it has been submitted to NCBI or not, and the biological item number has been received, which in my opinion is a necessary requirement before the manuscript is accepted or published.

Point 4 : Authors should check the entire manuscript for English and grammar improvements.

Author Response

Response

Dear Editor,

First, we would like to express our sincere appreciation to you and the reviewers for your precious time reviewing our manuscript. The suggestions made have greatly improved the quality of our manuscript. We have carefully revised our manuscript according to the comments. The revisions we made have been listed as follows, point by point.

Dear reviewers 3,

First of all, thank you for your precious time reviewing our manuscript. The opinions put forward are of great help to our article and significantly improve the quality of our manuscript. We have carefully revised our manuscript in response to comments. The details in the article have also been supplemented and revised. The changes we have made are listed one by one below.

Point 1:  Line 15- I. tectorum full name when writing first time.

Response 1:  Thank you very much for your valuable suggestions on our article. As the teacher said, the scientific name of a species should be expressed in full when it first appears. According to the teacher's suggestion, we have changed the scientific name of the species in line 15 to the full name.

Point 2 :  first is quality of figures are too poor it must should be improved.

Response 2 :  Thank you very much for your valuable suggestions on our article. As the teacher said, figures use a higher resolution graph better. We will uniformly change the whole text pictures into a TIF file and present the experimental data clearly and straightforwardly.

Point 3 : The second question is about the availability of the data, whether it has been submitted to NCBI or not, and the biological item number has been received, which in my opinion is a necessary requirement before the manuscript is accepted or published.

Response 3 : Thank you very much for your valuable suggestions on our articles. As the teacher said, the research data for this manuscript need to be provided before publishing this manuscript. All data in this study were stored on the Megi Bio platform (www.majorbio.com) and all data processing was implemented.

Point 4 : Authors should check the entire manuscript for English and grammar improvements.

Response 4 : Thank you very much for your valuable suggestions on our articles. We apologize for the poor spelling, grammar and syntax of the article. Due to these issues, we performed a detailed review of the revised manuscript to improve English and grammatical expression throughout the text.

Round 2

Reviewer 1 Report

The manuscript can be accepted in present form.